# A mapping review of methicillin-resistant *Staphylococcus aureus* proportions, genetic diversity, and antimicrobial resistance patterns in Cameroon

Nene Kaah Keneh[1,2], Sebastien Kenmoe[1], Arnol Bowo-Ngandji[3], Jane-Francis Tatah Kihla Akoachere[1], Hortense Gonsu Kamga[4], Roland Ndip Ndip[1], Jean Thierry Ebogo-Belobo[5], Cyprien Kengne-Ndé[6], Donatien Serge Mbaga[3], Nicholas Tendongfor[7], Lucy Mande Ndip[1,2], Seraphine Nkie Esemu[1,2]*

1 Department of Microbiology and Parasitology, University of Buea, Buea, Cameroon, 2 Laboratory for Emerging Infectious Diseases, University of Buea, Buea, South West Region, Cameroon, 3 Department of Microbiology, The University of Yaounde I, Yaounde, Cameroon, 4 Faculty of Medicine and Biomedical Sciences, The University of Yaounde I, Yaoundé, Cameroon, 5 Center for Research in Health and Priority Pathologies, Institute of Medical Research and Medicinal Plants Studies, Yaounde, Cameroon, 6 Epidemiological Surveillance, Evaluation and Research Unit, National AIDS Control Committee, Douala, Cameroon, 7 Department of Public Health and Hygiene, University of Buea, Buea, Cameroon

* esemu2003@yahoo.co.uk

## Abstract

### Background

The emergence of methicillin-resistant *Staphylococcus aureus* (MRSA) has increased and poses a significant threat to human and animal health in Cameroon and the world at large. MRSA strains have infiltrated various settings, including hospitals, communities, and live-stock, contributing to increased morbidity, treatment costs, and mortality. This evidence synthesis aims to understand MRSA prevalence, resistance patterns, and genetic characterization in Cameroon.

### Methods

The methodology was consistent with the PRISMA 2020 guidelines. Studies of any design containing scientific data on MRSA prevalence, genetic diversity, and antimicrobial resistance patterns in Cameroon were eligible for inclusion, with no restrictions on language or publication date. The search involved a comprehensive search strategy in several databases including Medline, Embase, Global Health, Web of Science, African Index Medicus, and African Journal Online. The risk of bias in the included studies was assessed using the Hoy et al tool, and the results were synthesized and presented in narrative synthesis and/or tables and graphs.

### Results

The systematic review analyzed 24 studies, mostly conducted after 2010, in various settings in Cameroon. The studies, characterized by moderate to low bias, revealed a wide

**Data Availability Statement:** All relevant data are within the paper and its Supporting Information files.

**Funding:** The author(s) received no specific funding for this work.

**Competing interests:** The authors have declared that no competing interests exist.

prevalence of MRSA ranging from 1.9% to 46.8%, with considerable variation based on demographic and environmental factors. Animal (0.2%), food (3.2% to 15.4%), and environmental samples (0.0% to 34.6%) also showed a varied prevalence of MRSA. The genetic diversity of MRSA was heterogeneous, with different virulence gene profiles and clonal lineages identified in various populations and sample types. Antimicrobial resistance rates showed great variability in the different regions of Cameroon, with notable antibiotic resistance recorded for the beta-lactam, fluoroquinolone, glycopeptide, lincosamide, and macrolide families.

## Conclusion

This study highlights the significant variability in MRSA prevalence, genetic diversity, and antimicrobial resistance patterns in Cameroon, and emphasizes the pressing need for comprehensive antimicrobial stewardship strategies in the country.

## Introduction

Methicillin-resistant *Staphylococcus aureus* (MRSA) has become a major global health problem, causing diseases ranging from skin infections to septic shock. The MRSA infections are generally treatable, though the infections have a 30-day mortality rate of 20–40% [1]. Over time, penicillin introduced in 1940 was used to treat *Staphylococcus aureus* infections until the bacterium developed resistance leading to the development of semi-synthetic penicillins such as methicillin [2]. Methicillin-resistant strains of *S. aureus* (MRSA) rapidly emerged and became a significant clinical problem in European hospitals during the 1960s, and in the United States and other regions during the 1970s [3]. MRSA has been continuously expanding geographically, with reports in many countries around the world and in Africa, including Cameroon [4, 5]. The development of MRSA led to increased morbidity, treatment costs, and mortality [6]. Since the 1990s, new strains of MRSA have emerged, expressing different types of resistance genes responsible for infections occurring in healthy people in communities and farms without hospital contact. Depending on the type of genotype expressed, MRSA can be classified into three main categories: hospital-associated MRSA (HA-MRSA), community-associated MRSA (CA-MRSA), and livestock-associated MRSA (LA-MRSA) [3]. CA-MRSA strains are generally susceptible to a variety of non-beta-lactam antibiotics, resistant to beta-lactams, and commonly possess SCCmec types IV and V and the Panton-Valentin leukocidin (PVL) gene. On the other hand, HA-MRSA strains, are resistant not only to beta-lactams, but also to other types of antibiotics, and are mainly associated with SCCmec types I, II, and III [7, 8].

MRSA frequently colonizes the human nose, throat, skin, and gastrointestinal tract, and more than 50% of colonized individuals develop the disease [3]. In addition to globalization facilitated by increased travel and trade, the spread of MRSA strains in community and hospital settings worldwide could also be facilitated by demographic (age, gender, region, poverty), environmental (poor sanitation, waste runoff from intensive agriculture, inadequate housing, poor water supply, crowded or medical environments, environmental contamination, and geographic movement of infected humans and animals) and host factors (immune status and antimicrobial use) [9–11].

Worldwide, MRSA infection rates vary year to year. According to National Nosocomial Infection Surveillance in the USA, MRSA-causing nosocomial infection increased from about

2.0% in 1974 to 22.0% in 1995 and then to 63.0% in 2004 [9–11]. In Taiwan, the MRSA infection rate increased dramatically from 9.8% in 1999–2000 to 56% in 2004–2005 [9–11]. In Africa, MRSA was reported as early as 1978 when a hospital outbreak occurred in 1986–1987 in South Africa [12]. Generally, most African countries have shown prevalence rates of MRSA ranging from 25% to 50% [4]. In the early 2000s, studies conducted on HA-MRSA in South Africa, Kenya, Nigeria, and Cameroon reported a prevalence ranging from 21 to 33.3% and a prevalence of below 10% in Tunisia, Malta, and Algeria [13]. In Cameroon, published studies reveal a wide range of MRSA prevalence in various settings in humans, animals, and environment [4].

MRSA is an ecological problem that impacts human, animal, and environmental health. Addressing this problem requires a clear knowledge of its prevalence in the healthy population, environment, and animals at the national level to support effective 'One Health' prevention and control strategies [14]. Systematic reviews are exhaustive and consider every bit of evidence on the topic, a few on MRSA have been carried out in some countries in the world [4, 5] but in Cameroon none to our knowledge. Monitoring and surveillance of MRSA circulation in hospitals, communities, animals, and environments is a critical aspect of public health for all health systems. This study is intended to help develop strategies for public health interventions to reduce MRSA infection in Cameroon. This is done through this systematic review to answer the question what is the prevalence, antibiotic resistance patterns, and genetic characterization of MRSA in Cameroon?

## Materials and methods

### Search strategy

This systematic review was conducted according to the Preferred Reporting Items for Systematic Reviews and Meta-Analyses (PRISMA 2020) guidelines (S1 Table) [15]. A systematic search was performed on the 6<sup>th</sup> of April 2023 in four major databases (Medline (Ovid), Embase (Ovid), Global Health (Ovid), Web of Science, African Index Medicus, and African Journal Online), which are recommended for their extensive coverage in systematic reviews (S2 Table) [16]. The search strategy used a combination of keywords and medical subject headings (MeSH) related to MRSA and Cameroon (S2 Table). In addition to the database search strategy, a manual search was performed by examining reference lists of all included studies, relevant articles in the field, and previous regional systematic reviews. Studies were included in our review if they were conducted in Cameroon and presented data on MRSA proportions, genetic diversity, and antimicrobial resistance patterns, regardless of study design, published in any language, and without publication date restrictions. We excluded studies that did not report on MRSA or that were not conducted in Cameroon.

### Study selection

Titles and abstracts of all sources were reviewed to exclude articles that clearly could not be included, and duplicate articles were removed using EndNote software (version X9, Clarivate Analytics). The selection process for this study was conducted using the Rayyan systematic review web-based tool [17]. Two reviewers independently screened the titles and abstracts of identified articles according to the eligibility criteria of the Rayyan platform [17]. Full-text articles were obtained for any study that met the inclusion criteria or for which the abstract did not contain sufficient information. In the second step, two reviewers independently reviewed the full-text articles to determine if they were eligible for inclusion in the study. Any discrepancies were resolved by discussion or consultation with a third reviewer. The selection process

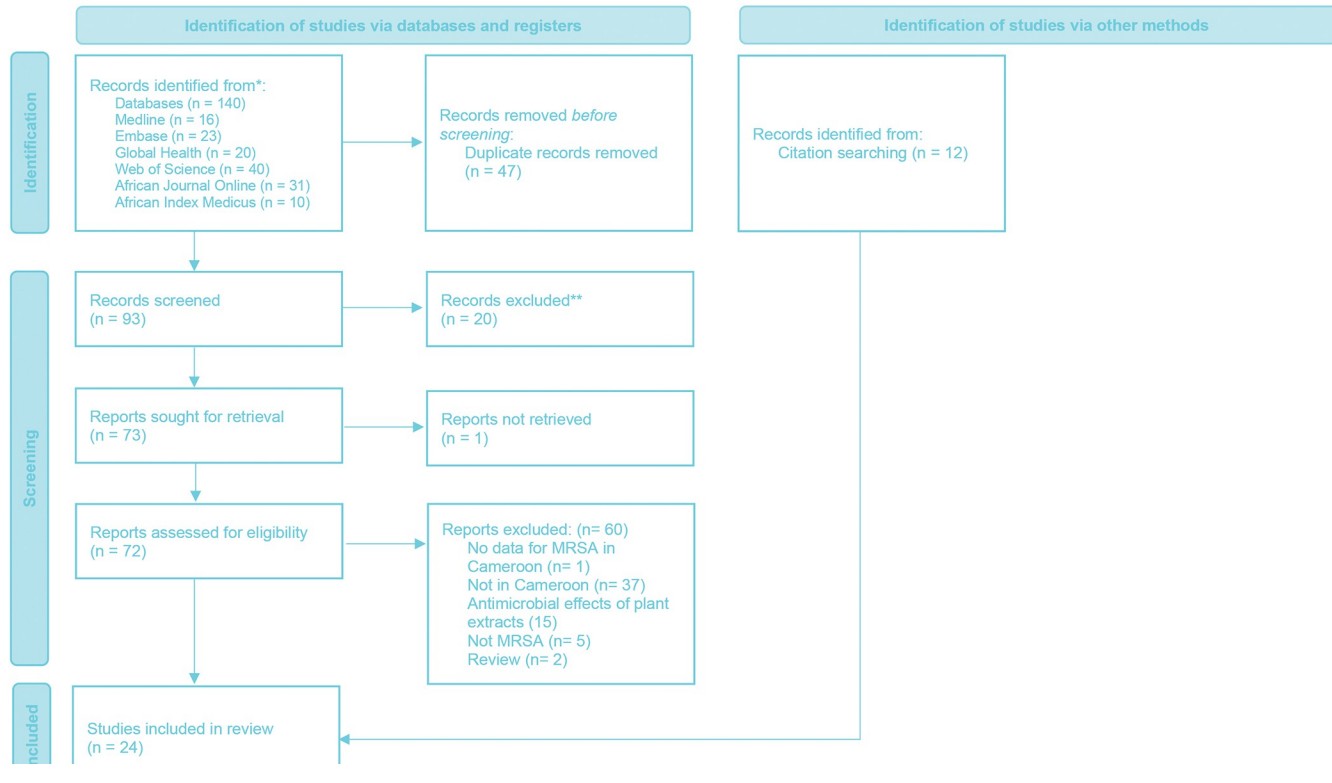

**Fig 1. Identification of included studies.**

was documented using a PRISMA flowchart detailing the number of studies identified, screened, and included or excluded, and the reasons for exclusion (Fig 1).

## Data extraction

We developed and tested an extraction template created using Google Forms. Two independent reviewers then evaluated the full text of the remaining studies for detailed data collection. The following data were extracted: Author, year of publication, study setting/sampling location, study period, number of cases studied, number of MRSA isolates, sources of isolates, MRSA identification test, sample types, number of participants, number of participants with MRSA, antibiotic resistance profile of MRSA isolates (antibiotic, number of antibiotic-resistant MRSA isolates), molecular typing methods, and number of MRSA types. Any disagreements were resolved by discussion or consultation with a third reviewer.

## Risk of bias assessment and data analysis

The quality assessment of the individual studies was performed using the tool of Hoy et al (S3 Table) [18]. Two authors independently assessed the quality of the included studies using a validated study quality assessment tool. The tool assesses external and internal validity. Any discrepancies in the assessment of the risk of bias were resolved by discussion or consultation with a third reviewer. Data were synthesized and presented using a narrative synthesis and/or tables and graphs.

## Results

### Study selection

The process of identifying relevant articles for inclusion in this study is represented graphically in Fig 1. A total of 140 articles were found in the databases, of which 93 titles/abstracts were reviewed and 72 complete reports evaluated. One report was not retrieved, and 20 reports were excluded based on predefined criteria. A total of 60 identified articles were excluded for various reasons. Finally, 24 studies were included in our review and met the criteria for MRSA proportions, genetic diversity, and antimicrobial resistance patterns in Cameroon [13, 19–41]. Among the included studies, there were 22 studies on human MRSA epidemiology [13, 20–24, 26–41], one study on animal epidemiology [25], two studies on food epidemiology [19, 21], one study on environmental epidemiology [38].

### Studies characteristics of the epidemiology of MRSA in Cameroon

The risk of bias was considered moderate in 15 studies and low in 9 studies (S4 Table). The review included studies conducted in hospitals in 20 studies (83.3%) and community settings in 3 studies (12.5%) (S5 Table). Two studies (8.3%) were conducted in the outpatient setting, while three studies (12.5%) were conducted in both the inpatient and outpatient settings. The studies were conducted over a broad range of periods spanning several decades. They began as early as 1996–1997 and extended up to more recent in 2021. The review included studies conducted in urban settings in 13 studies (54.1%), rural settings in five studies (20.8%), and two studies (8.3%) conducted in mixed settings. Most studies were conducted in recent years, with 22 studies conducted after 2010. The review covers 10 regions of Cameroon, with the Central region having the highest frequency with 11 (36.67%) publications. Of the 24 studies included, 8 were conducted in the Central Region (33.3%) [13, 22, 24, 27, 37], and Five in the Western Region (20.8%) [30, 32, 35, 36]. The Littoral (8.3%) [20, 41] and South West (8.3%) regions [21, 38] were studied twice each, while one study covered the Centre, Littoral, and South West regions (4.1%) [26]. In addition, one study included the Central and Southwest regions (4.1%) [39], and another focused on the Northwest and Southwest regions (4.1%) [19]. One study included all regions (4.1%) [33], and another covered the Adamawa and Far North regions (4.1%) [34]. Finally, two studies did not contain clear regional information [25, 40]. The age of the study population varied, ranging from 1 month to 92 years, with adult studies being the most studied population category (7 studies). Studies included hospital and community-based research, with some focusing on specific populations such as healthcare personnel, and patients with specific disease conditions; animals, foods, and/or environmental samples. These studies used a variety of methods to identify MRSA, including culture, PCR, and disc diffusion techniques. The types of samples collected in these studies were diverse, including ear, pus, throat, urine, genital swabs, blood, feces, and environmental samples.

### Prevalence of MRSA in humans

The prevalence of MRSA in humans across 19 studies demonstrated its variability within different populations, settings, and periods [13, 20–24, 26–28, 30–39] (Fig 2 and S6 Table). Only one community-based study in Buea found a 5.7% prevalence of MRSA in asymptomatic patients [21]. Almost all reports were hospital-based studies indicating a prevalence ranging from 1.9% among HIV/AIDS patients to 46.8% among patient with various diseases [28, 39]. Healthcare workers had a prevalence of 3.1%, which surged to 45.0% when considering among S. aureus isolates [22, 23]. Patients with specific diseases such as diabetes mellitus showed a prevalence of 38.9%, while those with HIV/AIDS ranged from 1.9% to 33.3% [22, 28]. Patients

| Study | Positive | Total | | Events | 95%−CI |
|---|---|---|---|---|---|
| **Adamawa** | | | | | |
| Mohamadou et al., 2022_Patients with various diseases (S. aureus positives) | 52 | 201 | | 25.87 | [19.97; 32.50] |
| **Centre** | | | | | |
| Eyoh et al., 2013_Healthcare workers | 8 | 254 | | 3.15 | [ 1.37; 6.11] |
| Eyoh et al., 2021_Healthcare workers (S. aureus positives) | 9 | 20 | | 45.00 | [23.06; 68.47] |
| Eyoh et al., 2021_Patients with diabetes mellitus (S. aureus positives) | 7 | 18 | | 38.89 | [17.30; 64.25] |
| Eyoh et al., 2021_Patients with HIV/AIDS (S. aureus positives) | 19 | 57 | | 33.33 | [21.40; 47.06] |
| Foloum et al., 2021_Clinically ill and asymptomatic patients | 47 | 1683 | | 2.79 | [ 2.06; 3.70] |
| Gonsu et al., 2020_Patients with various diseases | 17 | 127 | | 13.39 | [ 8.00; 20.56] |
| Kengne et al., 2020_Patients with HIV/AIDS | 1 | 53 | | 1.89 | [ 0.05; 10.07] |
| Njoungang et al., 2015_Unclear | 28 | 201 | | 13.93 | [ 9.46; 19.50] |
| **Far North** | | | | | |
| Mohamadou et al., 2022_Patients with various diseases (S. aureus positives) | 40 | 179 | | 22.35 | [16.47; 29.16] |
| **Littoral** | | | | | |
| Bissong et al., 2016_Patients with skin and soft tissue infections | 15 | 114 | | 13.16 | [ 7.56; 20.77] |
| **Multiregions** | | | | | |
| Gonsu et al., 2013_Healthcare workers; Patients with various diseases | 102 | 295 | | 34.58 | [29.16; 40.31] |
| Massongo et al., 2021_Positives isolates | 412 | 10218 | | 4.03 | [ 3.66; 4.43] |
| Sinda et al., 2020_Patients with various diseases | 162 | 346 | | 46.82 | [41.47; 52.23] |
| **South West** | | | | | |
| Esemu et al., 2021_Asymptomatic patients | 3 | 52 | | 5.77 | [ 1.21; 15.95] |
| Nkwelang et al., 2009_Patients with wounds and health personnel (finger nails and nostrils) | 80 | 231 | | 34.63 | [28.51; 41.15] |
| **West** | | | | | |
| Kesah et al., 2013_Patients with various diseases (S. aureus positives) | 43 | 100 | | 43.00 | [33.14; 53.29] |
| Manhafo et al., 2021_Patients with suppurating wounds or abscesses | 9 | 52 | | 17.31 | [ 8.23; 30.33] |
| Marbou et al., 2020_Clinically ill (Metabolic syndrome) and asymptomatic patients | 91 | 604 | | 15.07 | [12.31; 18.17] |
| Nankam et al., 2021_Patients with various diseases | 7 | 52 | | 13.46 | [ 5.59; 25.79] |
| Ngalani et al., 2020_Pregnant and non pregnant | 53 | 129 | | 41.09 | [32.50; 50.09] |

10  20  30  40  50  60

**Fig 2. Variability in MRSA prevalence across different populations, settings, periods, and geographical locations in Cameroon.**

with skin and soft tissue infections had a prevalence of 13.2% [20], and those with suppurating wounds or abscesses reported 17.3% [31]. A broad spectrum was observed among patients with various diseases, with rates spanning from 13.4% to 46.8% [27, 30, 34, 35, 39]. Pregnant and non-pregnant individuals presented a rate of 41.1% [36], while mixed populations presented diverse prevalence ranging from 2.8% to 34.6% [24, 26, 32, 38]. Out-patient settings had exhibited a rising prevalence, starting from 15.1% and peaking at 45.0% [22, 32]. For in-patients, the traumatology unit showed a prevalence of 17.3% [31], while a higher prevalence of 34.6% was observed in general in-patient wards [26]. The surgical department of in-patient settings had a lower prevalence at 13.5% [35], like the intensive care units which stayed at 13.4% [27]. Mixed settings, which have encompassed various departments, displayed a diverse prevalence, ranging from 2.8% to 46.8% [20, 24, 37, 39]. A subgroups data by Sinda et al. (2020) showcased variance by gender, age, ward type, and length of hospitalization, drawing attention to a 59.7% prevalence in surgical units [39]. MRSA prevalence demonstrated marked variability across locations (Fig 3). In rural areas encompassing towns such as Bangangte, Mbouda, Dschang, and Bafang, prevalence ranged from 13.5% in Bangangte to 41.1% in Bafang [30–32, 35, 36]. Urban areas, especially in cities like Yaounde, Buea, Douala, and Limbe, demonstrated a diverse prevalence spectrum from 1.9% to 45.0% [23, 28, 38, 39].

The basemap was taken from https://www.naturalearthdata.com and modified with QGIS software version 3.16.0-Hannover. We included 24 studies and for each study, we considered the outcomes as distinct reports, focusing on the prevalence in humans, prevalence in animals,

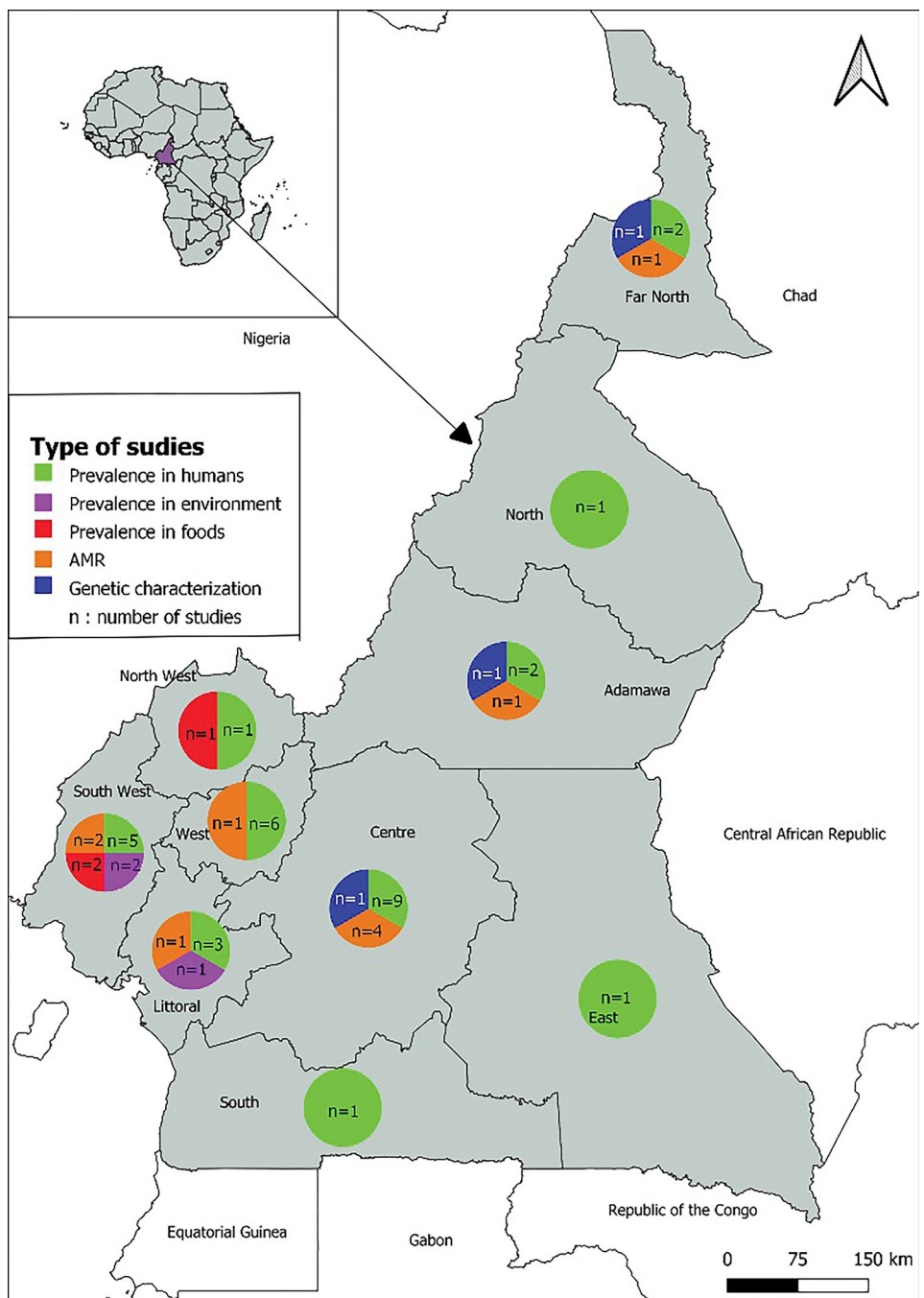

**Fig 3. Map of the Cameroon showing study locations and the number of report used in the review.**

prevalence in the environment, prevalence in food, profile of antimicrobial resistance of MRSA strains, or genetic characterization of MRSA strains. For studies spanning multiple regions or nationally, we counted the outcomes from each region as a separate report. This yielded a total of 53 reports: 31 on the prevalence in humans, 1 on the prevalence in animals, 3 on the prevalence in the environment, 3 on the prevalence in food, 10 on the profile of antimicrobial resistance of MRSA strains, and 5 on the genetic characterization of MRSA strains.

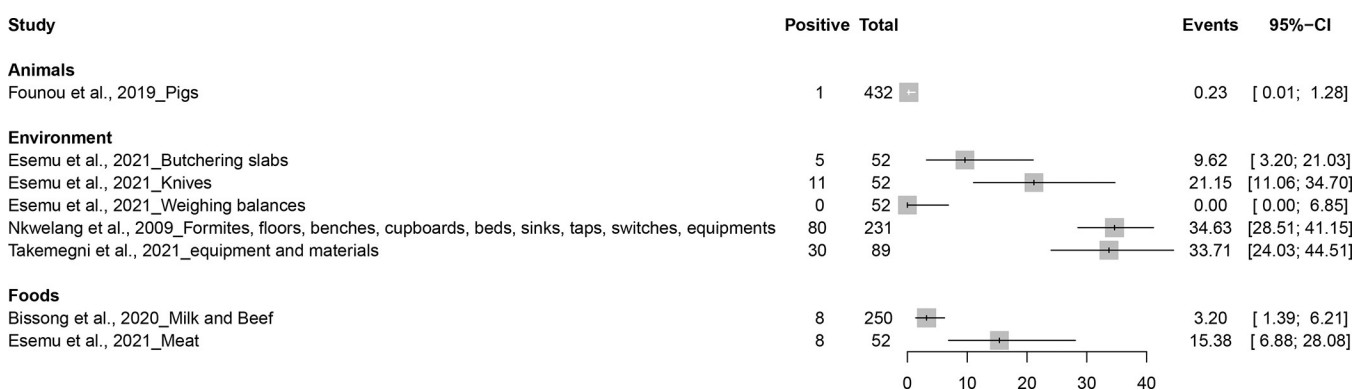

**Fig 4. MRSA prevalence in animals, foods, and different environmental samples in Cameroon.**

There were 2 studies on the prevalence in animals and the molecular characterization of MRSA strains that did not specify the study region.

## Prevalence of MRSA in animals, foods, and environment

MRSA prevalence varied in different enviroments with a high limit of 34.63% which was observed in Nkwelang et al (2009) a study carried out in Buea from hospital-based environmental samples [38]. The least prevalence (0.2%) was observed in animals in the study carried out in pigs from three abattoirs/markets by Founou et al. (2019) [25]. Bissong et al. (2020) reported a quite low prevalence of 3.2% in a community-based study in mixed settings in Bamenda, Buea, and Kumbo on milk and beef samples [19] (Fig 4 and S7 Table).

## Antimicrobial resistance patterns of MRSA isolates

Rates of antimicrobial resistance in MRSA vary considerably among studies (Fig 5 and S8 Table). Aminoglycoside resistance was reported in several studies. For gentamicin, resistance rates ranged from 22.2% in Dschang [31] to 55.4% in an unspecified location [34]. Resistance to amikacin was relatively low, with no resistance observed in Douala and Buea, and a 42.9% resistance rate in Yaounde [20, 21, 31, 34, 37]. Other aminoglycosides, such as kanamycin and tobramycin, showed higher resistance rates of 69.7% and 72.5%, respectively, in Yaounde [29]. The resistance rate to netilmycin in Yaounde was 32.1% [37].

Antimicrobial resistance patterns of MRSA to beta-lactam cephalosporins revealed high rates of resistance. Resistance to cefoxitin was consistently observed in studies conducted in Yaounde [29, 37] and the Adamawa and Far North regions [34], with a 100% resistance rate among MRSA isolates [29, 34, 37]. Similarly, a 100% resistance rate to cefepime was reported in Buea [21]. Resistance to ceftriaxone was also high, with a resistance rate of 90.5% in a study conducted in Buea, Yaounde, and Limbe [39].

Resistance to beta-lactam-penicillins has shown variable proportions across cities and populations. In a 2016 study by Bissong et al. in Douala on patients with skin and soft tissue infections, resistance rates were 40.0% for amoxicillin, 80.0% for penicillin, and 100% for oxacillin [19]. Another study by Esemu et al (2021), conducted in Buea, reported 100% Ampicillin resistance in asymptomatic patients and various environmental and food samples [21]. A study conducted in Dschang by Manhafo et al (2021), focusing on patients with suppurating wounds or abscesses, found resistance rates of 11.1% for oxacillin, 44.4% for penicillin and 77.8% for amoxicillin [31]. The 2022 study by Mohamadou et al. found resistance rates of 46.7% for amoxicillin-clavulanate, 89.1% for penicillin and 100% for oxacillin [34]. The 2015 study by

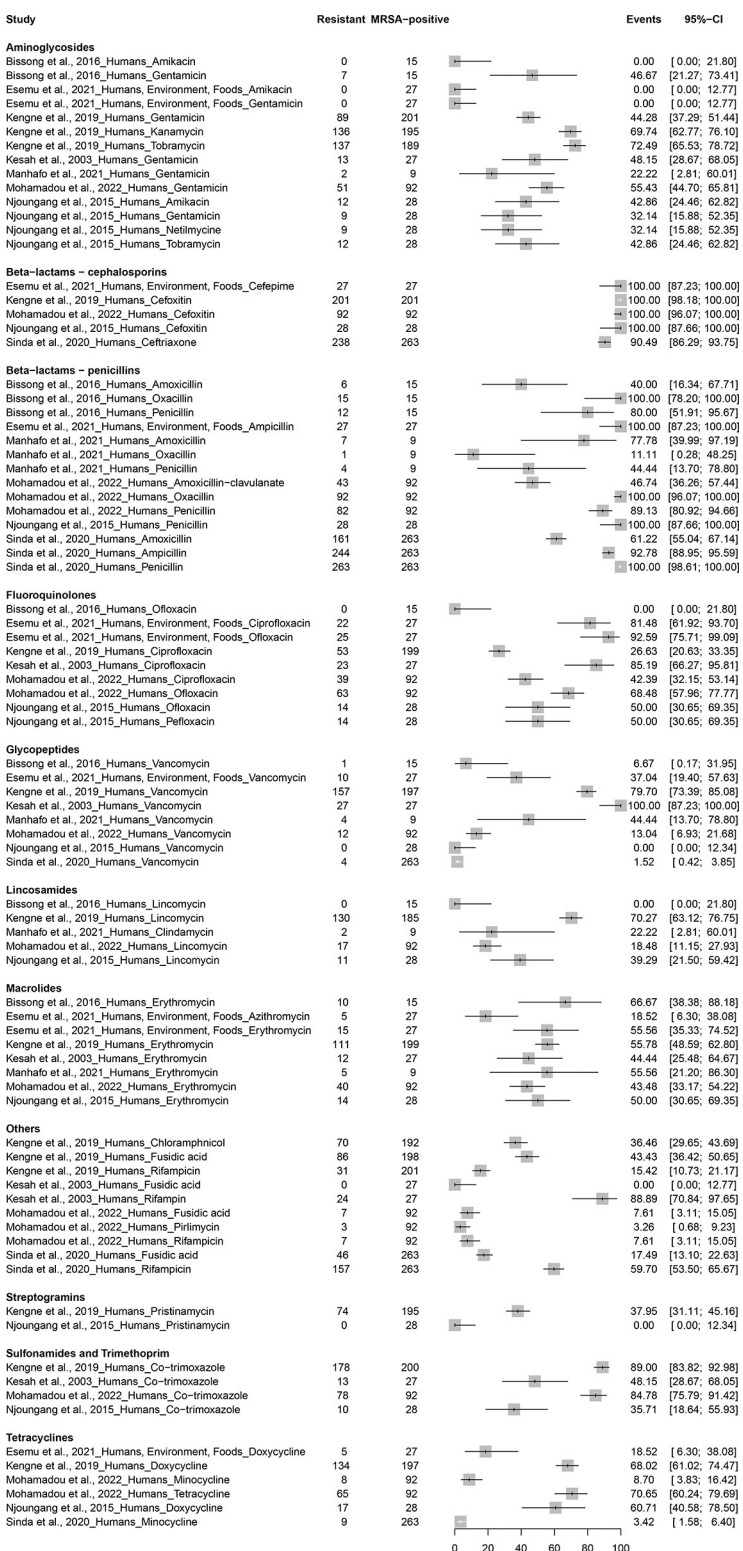

**Fig 5. Antimicrobial resistance patterns in MRSA across various regions and populations in Cameroon.**

Njoungang et al. in Yaounde reported 100% resistance to Penicillin, while Sinda et al. (2020), conducting research in Buea, Yaounde, and Limbe on patients with various diseases, recorded resistance rates of 61.2% for Amoxicillin, 92.8% for Ampicillin, and 100% for Penicillin [37, 39].

Fluoroquinolone resistance has shown significant variation between different cities and populations. Bissong et al. (2016), studying patients with skin and soft tissue infections in Douala, observed no resistance (0.0%) to Ofloxacin [20]. Esemu et al (2021), conducting research in Buea among humans, environmental samples, and food, found resistance rates of 81.5% for Ciprofloxacin and 92.6% for Ofloxacin [21]. In Yaounde, Kengne et al (2019) reported a 26.6% resistance rate to Ciprofloxacin in patients with various diseases, while Kesah et al (2003) observed a higher resistance rate of 85.2% to the same antibiotic [13, 29]. Mohamadou et al (2022) found resistance rates of 42.4% for ciprofloxacin and 68.5% for ofloxacin [34]. Njoungang et al (2015) in Yaounde reported resistance rates of 50.0% for Ofloxacin and Pefloxacin [37].

Glycopeptide resistance, particularly to vancomycin, has shown significant variation between different cities and populations. In a 2016 study by Bissong et al. in Douala on patients with skin and soft tissue infections, the resistance rate was 6.7% [20]. Another study by Esemu et al. (2021), conducted in Buea, reported a resistance rate of 37.0% in asymptomatic patients and in various environmental and food samples [21]. A study conducted in Yaounde by Kengne et al (2019) showed a 79.7% resistance rate, while Kesah et al (2003) reported a 100% resistance rate to Vancomycin in the same city [13, 29]. In Dschang, Manhafo et al (2021) found a resistance rate of 44.4% [31]. The study by Mohamadou et al. (2022) showed a resistance rate of 13.0% [34]. The 2015 study by Njoungang et al. in Yaounde reported no resistance (0.0%), while Sinda et al. (2020), conducting research in Buea, Yaounde and Limbe on patients with various diseases, recorded a low resistance rate of 1.5% [37, 39].

Lincosamide resistance has shown considerable regional and population variation. The 2016 study by Bissong et al. on patients with skin and soft tissue infections in Douala found no resistance (0.0%) to lincomycin [20]. In contrast, the 2019 study by Kengne et al. on patients with various diseases in Yaounde found a significantly higher resistance rate of 70.3% to the same antibiotic [29]. Manhafo et al (2021) reported a 22.2% resistance rate to clindamycin in patients with suppurating wounds or abscesses in Dschang [31]. The study by Mohamadou et al (2022) showed a resistance rate of 18.5% to Lincomycin [34]. A Lincomycin resistance rate of 39.3% was observed in the 2015 study by Njoungang et al. in Yaounde [37].

Antimicrobial resistance patterns of MRSA to macrolides in Cameroon showed distinct disparities among different populations and regions. In Douala, Bissong et al. (2016) found a high resistance rate of 66.7% to erythromycin in patients with skin and soft tissue infections [20]. In Buea, Esemu et al. (2021) reported a resistance rate of 18.5% to azithromycin and 55.6% to erythromycin in a mixed population of asymptomatic humans and various environmental and dietary factors [21]. Kengne et al (2019) observed an erythromycin resistance rate of 55.8% in patients with various diseases in Yaounde [29]. Similarly, Kesah et al (2003) reported a 44.4% resistance rate and Njoungang et al (2015) a 50.0% resistance rate to the same antibiotic in Yaounde [13, 37]. In Dschang, Manhafo et al. (2021) found a 55.6% resistance rate to erythromycin in patients with suppurating wounds or abscesses [31]. Finally, Mohamadou et al (2022) reported an erythromycin resistance rate of 43.5% [34].

Antimicrobial resistance patterns of MRSA to streptogramins, particularly pristinamycin, showed a notable contrast between two studies conducted in Yaounde, Cameroon. Kengne et al (2019) reported a resistance rate of 37.9% among a large population of patients with various diseases, indicating a considerable proportion of MRSA resistance [29]. However, a study by Njoungang et al. (2015) in the same city showed no resistance (0%) to pristinamycin in their population sample [37].

Antimicrobial resistance of MRSA to sulfonamides and trimethoprim, particularly cotri-moxazole, has shown substantial resistance rates in several studies in Cameroon. Kengne et al. (2019) reported a high resistance rate of 89.0% in patients with various diseases in Yaounde, closely followed by Mohamadou et al. (2022) with a resistance rate of 84.8% [29, 34]. The study by Kesah et al (2003) found a slightly lower resistance rate of 48.1% in patients with suppurating wounds or abscesses in Yaounde [13]. The lowest resistance was reported by Njoungang et al. (2015) with a rate of 35.7% in an unelucidated population [37].

MRSA resistance to tetracycline antibiotics in Cameroon was found to be variable across studies. Mohamadou et al. (2022) reported the highest rates of resistance to tetracycline (70.7%), followed closely by Kengne et al. (2019) who found a 68.0% resistance rate to doxycycline in various patients with diseases in Yaounde [29, 34]. Njoungang et al (2015) also reported a high rate of resistance to Doxycycline (60.7%) in Yaounde [37]. However, other studies such as Esemu et al. (2021) and Sinda et al. (2020) reported lower resistance rates of 18.5% and 3.4% to Doxycycline and Minocycline respectively [39, 21]. Mohamadou et al. (2022) also reported a lower resistance rate of 8.7% to Minocycline [34].

For MRSA resistance to other classes of antibiotics in Cameroon, resistance rates vary considerably. For example, fusidic acid showed resistance rates as high as 43.4% in a 2019 study by Kengne et al. in Yaounde, compared to lower resistance rates of 7.6% and 0% reported by Mohamadou et al. (2022) and Kesah et al. (2003) respectively [13, 29, 34]. Resistance to rifampicin also varied significantly, with a high resistance rate of 88.9% reported by Kesah et al. (2003), a moderate rate of 59.7% by Sinda et al. (2020), and a significantly lower rate of 15.4% and 7.6% by Kengne et al. (2019) and Mohamadou et al. (2022) respectively [13, 29, 34, 39]. The rate of resistance to chloramphenicol was 36.5% in the study by Kengne et al. (2019), while pirlimycin had a low resistance rate of 3.3% according to Mohamadou et al. (2022) [29, 34].

## Genetic diversity of MRSA isolates

Genetic characterization of MRSA revealed diverse virulence gene profiles in different populations and sample types (Fig 6 and S9 Table). Founou et al. (2019), using whole genome sequencing and MLST typing, detected the clonal lineage ST398 in a sample obtained from pigs between March and October 2016 [25]. In another study, Straus et al. (2015) applied the same sequencing and typing techniques to their human samples collected between 2005 and 2013 and identified the USA300 clonal lineage [40]. Eyoh et al (2021) reported rates of MRSA/LPV strains among health care personnel, patients with diabetes mellitus, and HIV-positive patients in Yaounde with 77.8%, 57.1%, and 31.6%, respectively [22]. Conversely, Mohamadou et al (2022) identified a rate of MRSA/PVL strains of 54.4% in patients with various pathologies [34]. Eyoh et al (2021) identified a low presence of type II SCCmec (5.7%) in healthcare workers, patients with diabetes mellitus, and HIV-positive patients in Yaounde [22]. Conversely, type IV SCCmec was significantly more present in these groups, with prevalences of 55.6% in healthcare workers, 71.4% in patients with diabetes, and 78.9% in HIV patients. SCCmec type V had a prevalence of 22.9% in the combined population groups. On the other hand, Mohamadou et al. (2022) found the following types of SCCmec in patients with various diseases: type I (23.9%), type II (7.6%), type III (14.1%), type IV (29.3%) and type V (22.8%) [34]. In addition, 2.2% of the isolates were untypable.

## Discussion

The review encompassed 24 mostly urban, hospital-based studies with Yaounde as the predominant city. These studies examined humans, animals, food, and environmental samples, with adults being the primary human category. A range of MRSA identification methods were

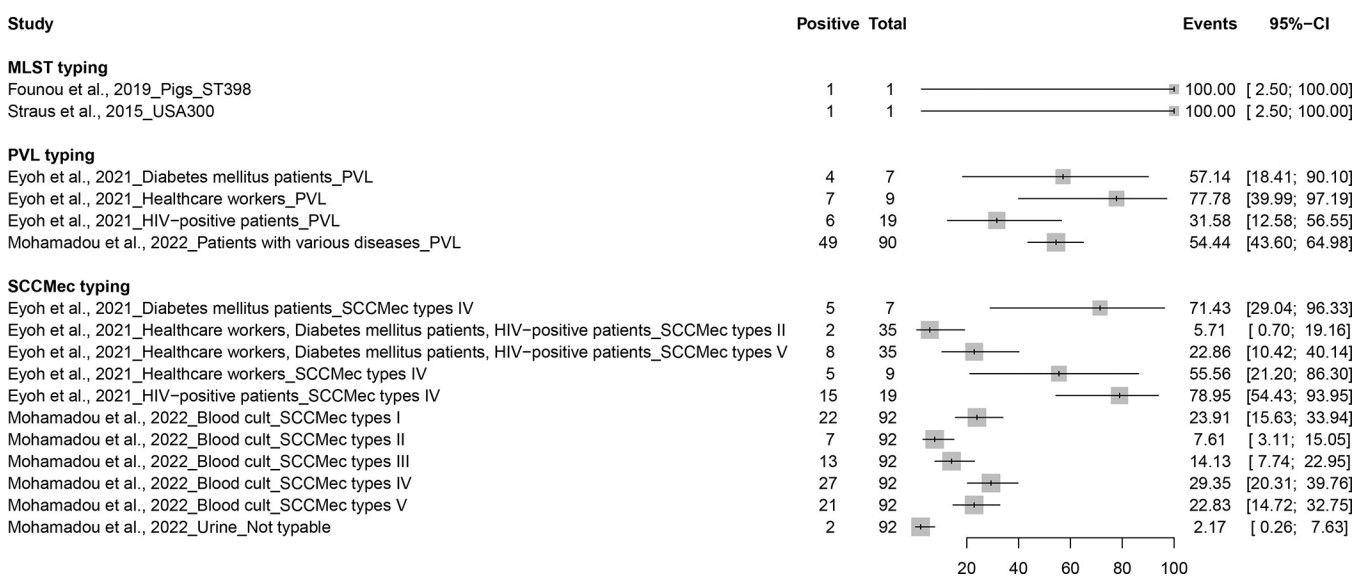

**Fig 6. Genetic diversity and virulence gene profiles of MRSA in different populations in Cameroon.**

applied to diverse sample types, revealing MRSA prevalence rates in humans from 1.9% to 46.8%. MRSA was also detected in food sources, with meat at 15.4% and milk and beef at 3.2%, and varied environmental sources, including a 34.6% in general environmental samples. MRSA strains exhibited varied resistance to several antibiotic families, with some resistance rates reaching up to 100%. Diverse MRSA clonal lineages were identified, notably the ST398 in pigs and USA300, and there was a wide distribution of SCCmec types among the studied populations.

## Prevalence of MRSA in humans

The results of this study, which reveal a wide range of MRSA prevalence (from 1.9 to 46.8%) in a broad spectrum of population groups in Cameroon, are in line with results from other African countries, placing the observed prevalence of MRSA within the documented range of 0–59% in a systematic review of MRSA nasal carriage in Africa [4]. This study highlights the complex interaction of various factors contributing to the significant variability in MRSA prevalence. These factors include demographic attributes such as gender, age, and settings, as well as sampling specifics such as sample size, location, and timing [42]. Laboratory procedures, including the type of test and sample, also play a role. In addition, the setting in which MRSA is encountered (hospital or community) adds a layer of complexity. In the hospital setting, factors such as intravascular catheterization, intensive care unit hospitalization, length of hospital stay, length of caregiver practice, and crowding can influence MRSA prevalence [43]. In addition, patient factors such as the presence of comorbidities, infection control practices including contact precautions, hand washing, sanitation, hygiene measures, education, and antibiotic use (frequent or infrequent, including the application of decolonization measures) all collectively contribute to the observed variability in MRSA prevalence [44]. The results of the study indicate a significant prevalence of MRSA in various population categories, which is a significant public health concern because of the potential of MRSA to cause serious infections and its resistance to many first-line antibiotics. Several factors may have contributed to this high prevalence. First, it is important to consider the wide range of population categories covered by the study, which included both healthy individuals and patients with various diseases [45]. The

high prevalence in patients with various illnesses and pregnant women is also of concern. These individuals often have weakened immune systems, making them more susceptible to infections, including those caused by MRSA [46, 47]. In addition, MRSA infections in these populations can be particularly difficult to treat because of the bacteria's resistance to many antibiotics. Another striking finding is the high prevalence of MRSA in individuals with isolates positive for multiple bacteria or S. aureus. This may indicate a higher microbial load or a weakened immune system in these individuals, both of which may increase the risk of MRSA infection [48]. This high prevalence of MRSA underscores the need for improved infection control measures, both in healthcare settings and in the broader community. Strategies could include better hygiene practices, appropriate use of antibiotics to reduce the development of resistance, and routine screening for MRSA in high-risk populations. Additional research is also needed to understand the factors contributing to the spread of MRSA and to develop more effective strategies for the prevention and treatment of MRSA infections.

## Prevalence of MRSA in foods

The results of the study suggest the presence of methicillin-resistant Staphylococcus aureus (MRSA) in food sources, namely meat and dairy products. This is a significant public health concern because food is a potential vector for transmission of MRSA to humans. The 15.4% prevalence of MRSA in meat is particularly notable. This relatively high rate can be attributed to several factors. One possible explanation is the use of antibiotics in animal husbandry, which can promote the development of antibiotic-resistant bacteria, including MRSA [49]. Animals raised for meat, such as poultry, pigs, and cattle, are often given antibiotics to prevent disease and promote growth. These practices can lead to the selection of antibiotic-resistant bacterial strains, which can then contaminate meat. The combined prevalence of MRSA in milk and beef (3.2%) is lower, but still a potential risk. Dairy cows can carry MRSA without showing symptoms, and MRSA can contaminate milk either directly from the udder or from the environment. Similarly, beef can become contaminated during slaughter and processing [50]. This highlights the need for good food handling and cooking practices to prevent MRSA infection. The presence of MRSA in food sources also highlights the importance of re-evaluating agricultural practices, particularly the use of antibiotics in animal agriculture. More prudent use of antibiotics, coupled with better husbandry and hygiene practices, could help reduce the prevalence of MRSA and other antibiotic-resistant bacteria in our food supply [51]. In addition, this raises questions about the potential for MRSA transmission through the food supply chain and the extent to which this transmission contributes to the overall burden of MRSA infections in humans. Additional research is needed to better understand these dynamics and to develop effective strategies to control MRSA in the food industry. Educating the public on proper food handling and preparation practices is also essential to minimize the risk of MRSA contamination. These include proper cooking of meats, washing hands and kitchen surfaces before and after handling raw meats, and avoiding cross-contamination between raw and cooked foods [52, 53].

## Prevalence of MRSA in environmental samples

The results of this study suggest a significant presence of methicillin-resistant Staphylococcus aureus (MRSA) in a variety of environmental sources, which may be a potential reservoir of MRSA and contribute to its transmission [54]. The discovery of MRSA on knives (21.2%) and butcher blocks (9.6%) is of concern, as these items are often used in food preparation, which can lead to contamination. This is particularly problematic in settings where the same knives and plates are used for multiple foods, without proper cleaning and disinfection in between.

This can contribute to the spread of MRSA in food and subsequently in humans who consume that food, which can lead to infections [54]. The high prevalence of MRSA in environmental samples (34.6%) and in hospital equipment and materials (33.7%) is also alarming. Environmental sources can serve as reservoirs for MRSA, facilitating its spread within communities and healthcare facilities. The high prevalence in hospital equipment is of particular concern, as it may contribute to healthcare-associated MRSA infections, which are often more difficult to treat due to the high level of antibiotic resistance in these strains [55]. It also implies the absence or ineffectiveness of infection control measures in these healthcare settings. The fact that no MRSA was found on the scales is a positive finding but should not lead to complacency. It suggests that regularly cleaned and disinfected surfaces, such as scales, are less likely to harbor MRSA. These results underscore the importance of rigorous sanitation and infection control measures in a variety of settings. Regular and effective cleaning of food preparation surfaces and utensils can reduce the likelihood of MRSA contamination and transmission in the food industry. In hospitals, strict disinfection protocols and proper handling of equipment and materials are essential to minimize the risk of MRSA transmission. Further research is needed to better understand the role of various environmental sources in MRSA transmission and to develop effective strategies to prevent the spread of this pathogen [56]. The use of molecular typing techniques could provide additional information on the origins and transmission routes of MRSA strains found in these environments.

## Livestock-associated methicillin-resistant Staphylococcus aureus

The detection of the ST398 clonal lineage of livestock-associated methicillin-resistant Staphylococcus aureus (LA-MRSA) in pigs, as shown in this study, is an important finding. LA-MRSA ST398 is a specific strain of MRSA that has been associated with livestock and has raised concerns because of its potential for zoonotic transmission, meaning it can be transferred between animals and humans. The existence of ST398 in pigs suggests that livestock may serve as a significant reservoir for this specific strain of MRSA. This is especially important for people who work closely with these animals, such as farmers and veterinarians, as they are at increased risk of contracting and spreading this bacteria [57]. In addition, if these animals enter the food chain, it could lead to contamination of meat products and contribute to the spread of this strain among consumers. This finding is part of a growing body of research indicating that livestock, and pigs, may be a significant reservoir of MRSA. However, the implications of this phenomenon are not yet fully understood. Although some studies suggest that LA-MRSA may not be as virulent or transmissible to humans as other strains, it is still resistant to several antibiotics, which can complicate treatment if infection occurs [58]. The identification of LA-MRSA ST398 in pigs also underscores the need for effective infection control measures on farms, including prudent use of antibiotics. Overuse of antibiotics in agriculture has been implicated in the emergence and spread of antibiotic-resistant bacteria, including MRSA [59]. Therefore, strategies to reduce the use of antibiotics in animal husbandry could help mitigate the spread of resistant strains such as ST398 [60]. However, it is important to note that these results are from a single study. Further research is needed to confirm the prevalence and importance of ST398 in pigs and other livestock and to determine the most effective strategies to control its spread.

## Antimicrobial resistance patterns of MRSA strains

The range of antimicrobial resistance rates reported in this study demonstrates the significant variability in MRSA resistance to different types of antibiotics. These studies conducted in a resource-limited country highlighted the wide variability in MRSA antimicrobial resistance.

Overuse and misuse of antibiotics, especially in animal husbandry and for non-therapeutic purposes, coupled with easy acquisition of drugs for self-medication, may be key factors in the increase of resistance [61]. Poor healthcare and hygiene practices, including the transfer of resistance through food chains and healthcare providers, could also be considered contributing factors. MRSA is known to be resistant to many antibiotics, including those in the beta-lactam family such as methicillin and penicillin. This is reflected in the data, with resistance to beta-lactam cephalosporins and beta-lactam penicillins reaching up to 100% [62, 63]. The varying rates of resistance across antibiotic families underscore the complexity of treating MRSA infections. For example, while resistance to aminoglycosides ranged from 0 to 72.5%, resistance to fluoroquinolones ranged from 0 to 92.6%. This suggests that while some MRSA strains are susceptible to these antibiotics, others are highly resistant. Interestingly, the study found MRSA resistance rates to the glycopeptide antibiotic vancomycin ranging from 0% to 100%. Vancomycin is considered an antibiotic of last resort for MRSA infections because of its ability to treat resistant strains [64]. However, the emergence of vancomycin-resistant MRSA is of great concern. Although the upper limit of this range is likely due to a small subset of MRSA, this is an important finding that underscores the pressing need for ongoing surveillance and development of new antimicrobial agents [65]. The relatively lower rates of resistance to other classes of antibiotics such as streptogramins (pristinamycin; 0–37.9%) and tetracyclines (3.4–70.7%) suggest that these antibiotics may still be effective against some MRSA strains. However, the range of resistance still implies the existence of strains resistant to these antibiotics. These resistance rates also highlight the importance of antibiotic stewardship, which involves the careful use of antibiotics to prevent the development and spread of resistant bacteria. Overuse and misuse of antibiotics are key factors in antibiotic resistance [66]. Therefore, strategies to promote appropriate antibiotic use are essential to combat the rise of MRSA and other antibiotic-resistant bacteria. Finally, these findings highlight the need for ongoing surveillance of antibiotic resistance in MRSA. Regular monitoring can identify emerging resistance patterns and inform treatment guidelines. However, it is important to note that antibiotic susceptibility can vary by geographic location and over time, so local data should also be considered when selecting treatment for MRSA infections [67].

## Molecular characterization of MRSA strains

The results of this study demonstrate the genetic diversity of MRSA strains in different populations and environments, which is a critical component in understanding the epidemiology of MRSA infections. Multilocus sequence typing identified distinct clonal lineages of MRSA, such as the ST398 lineage in swine and the USA300 lineage. The ST398 lineage has been associated with cattle and has been of particular concern in areas with significant swine production. Lineage USA300 is one of the most prevalent community strains of MRSA in the United States, known to cause severe skin and soft tissue infections. Identification of these specific lineages can help understand the source of infection and implement appropriate infection control measures [68]. Detection of Panton-Valentine leukocidin (PVL) in MRSA strains provides a deeper understanding of the epidemiology of MRSA. PVL is a toxin associated with some strains of S. aureus, including some MRSA strains, and is often linked to more severe skin and soft tissue infections [69]. Reported variations in the prevalence of MRSA/PVL in different populations could indicate differences in the types of MRSA strains circulating in these groups or variations in the populations' exposure to these strains. Detection of different types of staphylococcal cassette chromosome mec (SCCmec) also provides important information about the origins and characteristics of MRSA strains. SCCmec is the genetic element that carries the methicillin resistance gene in MRSA. Different types of SCCmec (I-XIV) are associated with

different MRSA strains and have implications for the strain's resistance profile, ability to spread, and pathogenic potential [70]. The varied distribution of SCCmec types among different population groups may reflect differences in exposure to different MRSA strains. The presence of non-typeable isolates suggests that new or uncommon types of SCCmec are circulating in the population [71]. These non-typeable isolates could potentially represent new strains of MRSA that have not yet been fully characterized. Overall, these results highlight the genetic diversity of MRSA and underscore the need for ongoing surveillance and research to track the evolution and spread of different MRSA strains. This information can help guide infection control strategies and the development of new treatments.

## Study limitations

This study has several limitations that influence the interpretation of its results. The absence of the mecA gene test, the gold standard for MRSA confirmation, in many studies raises questions about the reliability of MRSA identification. Furthermore, the primary reliance on the disk diffusion method for resistance profiling, without delving into the minimum inhibitory concentration, may compromise the accuracy of deduced resistance levels. The variability across studies in terms of host, context (such as CA-MRSA, HA-MRSA, or LA-MRSA), clinical manifestation (be it colonization or infection), and MRSA detection techniques clouds definitive conclusions about MRSA's impact in Cameroon and impedes a meta-analysis. Moreover, the results' generalizability is potentially skewed, considering the underrepresentation of several of Cameroon's ten regions and the minimal inclusion of pediatric populations. These factors should be weighed carefully when considering the broader implications and potential biases of our findings.

## Study implications

The results of this study have several important implications. They underline the need for further prospective studies and the implementation of routine surveillance to facilitate population-based analyses. This would not only overcome the limitations identified but also guide effective patient management through systematic monitoring of the antibiotic susceptibility profile of MRSA strains. The study underscores the need for preventive measures and infection control strategies to combat MRSA's challenges in Cameroon. Beyond the general call for contact precautions, sterilization, and disinfection, it's crucial to emphasize concrete measures such as the implementation of rigorous hygiene practices, educational awareness programs, and robust compliance monitoring systems within healthcare facilities. Though the studies reviewed did not delve deep into these measures, they remain paramount. There is also an evident necessity for systematic genotyping of MRSA across various sources from humans and animals to food and the environment. Enhancing waste management protocols, strengthening infrastructure, optimizing the use of medical equipment, and investing in both the number and training of healthcare personnel should be prioritized. Policymakers should adopt strict MRSA national treatment guidelines, encourage the uptake of effective antibiotic stewardship programs, and promote a reinforced "One Health" approach. This approach would joined together a multidisciplinary network encompassing community members, healthcare professionals, zoologists, and botanists, fortified further by both national and international partnerships.

## Conclusion

This research underscores a concerning high prevalence of MRSA in Cameroon, revealing strains that not only are genetically diverse but also demonstrate alarming resistance to first

line used antibiotics. Such findings elevate the importance of improved antimicrobial steward-ship and intensified infection control measures for Cameroon's public health sector. The insights from this study emphasize the imperative for immediate, coordinated action both for the well-being of individuals in Cameroon and to prevent wider regional implications.

## Supporting information

**S1 Table. PRISMA 2020 checklist.**
(DOCX)

**S2 Table. Search strategy.**
(DOCX)

**S3 Table. Items for risk of bias assessment.**
(DOCX)

**S4 Table. Risk of bias assessment.**
(DOCX)

**S5 Table. Individual characteristics of included studies.**
(DOCX)

**S6 Table. Individual characteristics of MRSA prevalence studies in humans.**
(DOCX)

**S7 Table. Individual characteristics of MRSA prevalence studiesin animals, foods, and environment.**
(DOCX)

**S8 Table. Antimicrobial resistance patterns of MRSA isolates.**
(DOCX)

**S9 Table. Genetic diversity of MRSA isolates in Cameroon.**
(DOCX)

## Author Contributions

**Conceptualization:** Nene Kaah Keneh, Sebastien Kenmoe, Seraphine Nkie Esemu.

**Data curation:** Nene Kaah Keneh, Sebastien Kenmoe, Arnol Bowo-Ngandji, Jean Thierry Ebogo-Belobo, Cyprien Kengne-Ndé, Donatien Serge Mbaga.

**Formal analysis:** Sebastien Kenmoe, Cyprien Kengne-Ndé.

**Methodology:** Nene Kaah Keneh, Sebastien Kenmoe, Arnol Bowo-Ngandji, Jane-Francis Tatah Kihla Akoachere, Hortense Gonsu Kamga, Roland Ndip Ndip, Jean Thierry Ebogo-Belobo, Cyprien Kengne-Ndé, Donatien Serge Mbaga, Nicholas Tendongfor, Lucy Mande Ndip, Seraphine Nkie Esemu.

**Project administration:** Nene Kaah Keneh, Seraphine Nkie Esemu.

**Validation:** Nene Kaah Keneh, Hortense Gonsu Kamga, Roland Ndip Ndip, Jean Thierry Ebogo-Belobo, Cyprien Kengne-Ndé, Donatien Serge Mbaga, Nicholas Tendongfor, Lucy Mande Ndip, Seraphine Nkie Esemu.

**Writing – original draft:** Nene Kaah Keneh.

**Writing – review & editing:** Nene Kaah Keneh, Sebastien Kenmoe, Arnol Bowo-Ngandji, Jane-Francis Tatah Kihla Akoachere, Hortense Gonsu Kamga, Roland Ndip Ndip, Jean Thierry Ebogo-Belobo, Cyprien Kengne-Ndé, Donatien Serge Mbaga, Nicholas Tendong-for, Lucy Mande Ndip, Seraphine Nkie Esemu.

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
