## [Decision Letter · Decision Letter 0]

18 Sep 2023

PONE-D-23-27108A mapping review of methicillin-resistant Staphylococcus aureus proportions, genetic diversity, and antimicrobial resistance patterns in CameroonPLOS ONE

Dear Dr. Esemu

Thank you for submitting your manuscript to PLOS ONE. After careful consideration, we feel that it has merit but does not fully meet PLOS ONE’s publication criteria

We look forward to receiving your revised manuscript.

Kind regards,

Mabel Kamweli Aworh, DVM, MPH, PhD. FCVSN

Academic Editor

PLOS ONE

Journal Requirements:

Reviewers' comments:

Reviewer's Responses to Questions

**Comments to the Author**

1. Is the manuscript technically sound, and do the data support the conclusions?

Reviewer #1: Partly

Reviewer #2: Yes

Reviewer #3: Yes

Reviewer #4: Yes

2. Has the statistical analysis been performed appropriately and rigorously? 

Reviewer #1: N/A

Reviewer #2: Yes

Reviewer #3: Yes

Reviewer #4: Yes

3. Have the authors made all data underlying the findings in their manuscript fully available?

Reviewer #1: Yes

Reviewer #2: Yes

Reviewer #3: Yes

Reviewer #4: Yes

4. Is the manuscript presented in an intelligible fashion and written in standard English?

Reviewer #1: Yes

Reviewer #2: Yes

Reviewer #3: Yes

Reviewer #4: Yes

5. Review Comments to the Author

Reviewer #1: Thank you for submitting this Manuscript for peer review and publication. Please find below my comments.

Keywords: The Authors should consider also including the keywords in the Abstract section. Additionally, MRSA should be written in full.

Abstract

Background: The authors should include that this study should rework the background section to indicate that this study is an evidence synthesis.

Introduction

The introduction of this journal article discusses the significant global health problem posed by Methicillin-resistant Staphylococcus aureus (MRSA), which causes various diseases with a high 30-day mortality rate. It describes the historical context as well as the colonization of MRSA in humans, animals, and the environment essentially linking the study to a One Health Approach. Overall, the introduction effectively establishes the context and importance of the study, but there are areas for improvement.

Areas for Improvement

1. The study can benefit from additional studies describing the status of MRSA in Cameroun. Also, the authors could consider moving lines 86-87 to immediately after line 61, to introduce early the context of the present study to readers.

2. As this is an evidence synthesis, the authors should include:

a. The main question this study aims to answer

b. A rationale for choosing this type of method as appropriate

c. If there have been any previous systematic reviews on this topic

3. Was there a protocol for this study? If yes, was it registered? If yes, include the registration details.

4. Line 55: clarify “they” in the statement.

5. Line 56. S.aureus. The authors should write it out in full since this is the first time the word is being used in the article.

Materials and methods

Overall, this section appears comprehensive but could benefit from some additional details and/or clarifications.

Areas for Improvement

Search Strategy

1. A justification for the selection of the databases

2. An inclusion of the interphase for Web of Science

3. Any gray literature searched or reason for exclusion

4. Line 106: The authors should clarify “meta-analyses”, why the distinction from journals?

5. Line 104: Please refer the reader to the appropriate appendix where the keywords and MeSH terms can be found.

6. Line 110: The authors should clarify the “Cameroonian framework”

Study selection

1. Lines 112-114: The authors should clarify “ Rayyan systematic review website”. To my knowledge, Rayyan is a web-based tool that can assist researchers in conducting systematic reviews. Did the authors use their eligibility criteria or were there some pre-determined criteria in Rayann?

2. Was there any blinding done so as not to bias the review?

3. Line 119: Please refer the reader to the appropriate appendix where the PRISMA flowchart can be found can be found. Please write out the full meaning of PRISMA.

Data Extraction

1. Line 122-123 -: Removing duplicates should be part of study selection and not data extraction.

2. Did the authors use a standardized tool for extraction? If not, was any tool used? If yes, how was the tool developed?

Risk of bias assessment and data analysis

1. Can the authors include a justification for the rationale for assessing the quality of the included sources?

Results

This section is comprehensive and well-documented.

Areas for Improvement

1. Line 140: meta-analysis. Can the authors justify the use of the word?

2. Line 165: Is Adamawa the same as Adamaoua (Line 165 and Line 237)?

3. Line 170-171: Please rework the 2 sentences for improvement in clarity and flow.

4. Line 176- 217: Please rework this section to capture the interest of the reader. The evidence in this section needs to be synthesized. The authors can consider combining studies with similarities as opposed to listing out the prevalence of MRSA in individual studies. Also, consider the inclusion of a summary statement about the prevalence of MRSA at the beginning of the section e.g., Line 228 and Line 326.

Discussion

This is comprehensive and well-detailed.

Appendix 1: Registration and Protocol. Can the authors clarify where the registration and protocol information can be located under Methods?

Appendix 2: Include interface for Web of Science

Appendix 6: Can the authors include a description of the headers for the three “settings” to differentiate them from each other?

Reviewer #2: REVIEW OF MANUSCRIPT: PONE-D-23-27108

1. Overall, the manuscript appears to be well-structured, thorough, and provides valuable insights into the Methicillin-Resistant Staphylococcus Aureus (MRSA) situation in Cameroon. Data provided by the researchers is adequate, however, it would be of more benefit if the discussion section provided a more comprehensive analysis of the implications of the research findings.

2. Abstract

The abstract provides a clear and concise overview of the research conducted in this manuscript. It effectively introduces the problem of methicillin-resistant Staphylococcus aureus (MRSA) and its global health threat, with a particular focus on its prevalence, resistance patterns, and genetic characteristics in Cameroon.

a. Even though authors reviewed several papers with information listed in the methods, it is not clear what data was extracted from published work which was included in the study.

b. Line 43: “The genetic diversity of MRSA was significant,” kindly state the level of significance in the study.

3. Introduction:

The introduction effectively sets the stage for the research by discussing the global spread of MRSA and its presence in Cameroon, emphasizing the need for surveillance and interventions to curb its dissemination.

a. Include information on global and local prevalence of MRSA if any.

b. Line 63-66: include reference(s).

c. Line 75-76: include reference(s).

4. Methods:

The materials and methods section are comprehensive and well-structured. The PRISMA 2020 guidelines, which is a widely recognized approach for systematic reviews establishes the credibility and reliability of the research process. The search strategy, including the databases used and the date of the search, is clearly detailed. The inclusion and exclusion criteria for study selection are well-defined, and the use of the Rayyan systematic review website for study selection is a practical and efficient approach. The data extraction process is thorough, and the risk of bias assessment using the Hoy et al. tool adds rigor to the research.

a. Kindly state how you take care of duplicate papers.

b. Kindly represent the bibliographic search strategy in tabular format.

c. Line 118-120: Reference flowchart.

d. Line 132-136: What level of agreement by the authors/investigators was considered during selection process?

5. Result:

The results section is detailed having various subheadings that looked at characteristics of the selected studies, the prevalence of MRSA, antimicrobial resistance patterns of MRSA isolates and the genetic diversity of MRSA in Cameroon.

a. Kindly number the tables/figures appropriately as figures and appendix are used interchangeably. Also, it is important not to include all the information on the studies included in the paper. Key /relevant finding should be included as prose, and this would significantly reduce the word count of the paper.

b. Line 152: include the setting of the last study.

c. Kindly provide the summary measure of the various categories of data on the forest plot. E.g., Summary measure of prevalence of MRSA from all the studies included in the paper.

6. Discussion:

The discussion section of the manuscript provides a detailed overview of the study's findings regarding the prevalence and characteristics of MRSA in various settings in Cameroon. Overall, it offers valuable insights into the state of MRSA in the region and its implications for public health. However, there are some aspects that could benefit from clarification, refinement, or expansion:

a. Clarity and Organization: The discussion section covers a wide range of topics related to MRSA, including prevalence, antibiotic resistance, genetic diversity, and implications. While comprehensive, it could benefit from better organization and grouping of related information to enhance readability and comprehension. The use of subheadings would be very useful to enhance readability.

b. Limitations:

• Line 540-555: The section mentions several limitations of the study, but it's important to emphasize their impact on the interpretation of the results in regards of generalization and potential biases.

c. Recommendations:

• Line 555-564: The section mentions the need for preventive measures and infection control strategies but could elaborate more on practical recommendations such as healthcare facilities, policymakers, and individuals can take to address the challenges posed by MRSA in Cameroon.

d. Conclusion:

Conclude the discussion section by summarizing the key takeaways from the study, the implications for public health in Cameroon, and the broader significance of the findings. This can help readers grasp the main messages more easily.

7. References:

References are relevant to the topic and are recent papers, however 40% of your references are older than 5 years. Kindly include newer references with older references consisting of about 20% of your entire references.

Reviewer #3: Line 27: I suggest if you want to put a significance to Cameroun you may use the bottoms up approach starting with Cameroun and ending with world wide

Line 101: What method this you use for your date? mm/dd/year or dd/mm/year. to prevent such confusion you could write the month out in words

The manuscript is well written and lots of work has been put into it.

I would suggest, as the topic says mapping"... this requires more visual representations like comparing geo-locations,use of charts ,tables etc which would make the work more engaging and easier to grasp as it contains too much information.

Reviewer #4: General comments: The article is generally well written with a good flow. The abstract is a good summary of the work, the objective is clearly stated and the methods clearly describes how to achieve the objectives. The literature review gives enough information based on the objectives and key words. Results are in keeping with the objectives and in in chronologically order. The discussion also provides providing interpretation of the findings in the light of other studies. Limitations for the study are clearly stated and the conclusion is succinct with recommendations clearly state.

Line 42: Will be good to have the range of values for the prevalence in animal, food and environment mentioned as well.

Line 103: For ease of replicating this study, it will be good to specify the key words and medical subject headings used in this study.

Line 150: Will be helpful if you give a brief description of Cameroon in terms of the regions. This will enable the reader appreciate the geographical coverage of the studies. Also good to specify the range of time over which studies were conducted clearly.

Line 175: Would have been good to order the prevalence (by year or magnitude) to enable the reader appreciate the trend.

Line 345: The first 3 sections in the discussion section appear to be a repetition of the results. The discussion section should not be meant for repeating the results.

Line 351 & 373: The abstract and result sections have a prevalence of "1.8 to 46.8%", while the discussion section contains "1.9 to 46.8%". There is need for consistency here.

Line 378: Not all the factors mentioned here were well captured in the result section. There is a need to review this sections to make sure the factors referred to are described in the result section.

6. PLOS authors have the option to publish the peer review history of their article (what does this mean?). If published, this will include your full peer review and any attached files.

Reviewer #1: **Yes: **Adeola Ayo

Reviewer #2: No

Reviewer #3: **Yes: **Folashade Onatola Toye

Reviewer #4: **Yes: **Bola Biliaminu Lawal

---

## [Author Response · Author response to Decision Letter 0]

14 Nov 2023

5. Review Comments to the Author

Reviewer #1: Thank you for submitting this Manuscript for peer review and publication. Please find below my comments.

Keywords: The Authors should consider also including the keywords in the Abstract section. Additionally, MRSA should be written in full.

Authors: Thank you for this valuable feedback on our manuscript. We have now added the

the keywords in the Abstract section and MRSA have been written in full.

Abstract

Background: The authors should include that this study should rework the background section to indicate that this study is an evidence synthesis.

Authors: Thank you for this valuable feedback on our manuscript. We have now edited the

the objective in the Abstract section to include evidence synthesis.

Introduction

The introduction of this journal article discusses the significant global health problem posed by Methicillin-resistant Staphylococcus aureus (MRSA), which causes various diseases with a high 30-day mortality rate. It describes the historical context as well as the colonization of MRSA in humans, animals, and the environment essentially linking the study to a One Health Approach. Overall, the introduction effectively establishes the context and importance of the study, but there are areas for improvement.

Areas for Improvement

1. The study can benefit from additional studies describing the status of MRSA in Cameroun. Also, the authors could consider moving lines 86-87 to immediately after line 61, to introduce early the context of the present study to readers.

Authors: Thank you for this valuable feedback on our manuscript. The necessary adjustments have been made.

2. As this is an evidence synthesis, the authors should include:

a. The main question this study aims to answer

Authors: Thank you for this valuable feedback on our manuscript. The last paragraph of the introduction has been edited accordingly.

b. A rationale for choosing this type of method as appropriate

Authors: Thank you for this valuable feedback on our manuscript. The last paragraph of the introduction has the reason for the method.

c. If there have been any previous systematic reviews on this topic

Authors: Thank you for this valuable feedback on our manuscript. A few such studies have been mentioned and referenced in the introduction, but none targeted Cameroon only.

3. Was there a protocol for this study? If yes, was it registered? If yes, include the registration details.

Authors: Our study employed a mapping review approach. Due to the broad objectives associated with mapping reviews, primary registration platforms like PROSPERO do not allow their registration. Unlike systematic reviews, which have narrower and more specific objectives necessitating protocol registration, mapping reviews are generally not subjected to the same requirement.

4. Line 55: clarify “they” in the statement.

Authors: Thank you for this valuable feedback on our manuscript. The word they in the statement has been clarified

5. Line 56. S.aureus. The authors should write it out in full since this is the first time the word is being used in the article.

Authors: Thank you for this valuable feedback on our manuscript. S. aureus has now been written in full.

Materials and methods

Overall, this section appears comprehensive but could benefit from some additional details and/or clarifications.

Areas for Improvement

Search Strategy

1. A justification for the selection of the databases

Authors: Thank you for this valuable feedback on our manuscript. We edited as “A systematic search was performed on the 6th of April 2023 in four major databases (Medline (Ovid), Embase (Ovid), Global Health (Ovid), Web of Science, African Index Medicus, and African Journal Online), which are recommended for their extensive coverage in systematic reviews”

2. An inclusion of the interphase for Web of Science

Authors: The Web of Science interface is a web-based platform as the other databases with no specific additional names.

3. Any gray literature searched or reason for exclusion

Authors: We did not include grey literature in our primary search. However, we extensively reviewed the citations of eligible studies, which led us to identify and incorporate an additional 12 studies from sources outside the primary databases.

4. Line 106: The authors should clarify “meta-analyses”, why the distinction from journals?

Authors: Thank you for this valuable feedback on our manuscript. We edited as “In addition to the database search strategy, a manual search was performed by examining reference lists of all included studies, relevant articles in the field, and prvious regional systematic reviews.”

5. Line 104: Please refer the reader to the appropriate appendix where the keywords and MeSH terms can be found.

Authors: Thank you for this valuable feedback on our manuscript. The corresponding appendix have now been listed just after the sentence.

6. Line 110: The authors should clarify the “Cameroonian framework”

Authors: Thank you for this valuable feedback on our manuscript. We edited as “We excluded studies that did not report on MRSA or that were not conducted in Cameroon.”

Study selection

1. Lines 112-114: The authors should clarify “ Rayyan systematic review website”. To my knowledge, Rayyan is a web-based tool that can assist researchers in conducting systematic reviews. Did the authors use their eligibility criteria or were there some pre-determined criteria in Rayann?

Authors: Thank you for this valuable feedback on our manuscript. We edited as “The selection process for this study was conducted using the Rayyan systematic review web-based tool.” And added a reference.

2. Was there any blinding done so as not to bias the review?

Authors: Study selection, data extraction, and risk of bias assessment were assessed independently by two authors blinded, as indicated in the methodology.

3. Line 119: Please refer the reader to the appropriate appendix where the PRISMA flowchart can be found can be found. Please write out the full meaning of PRISMA.

Authors: Thank you for this valuable feedback on our manuscript. The PRISMA flowchart has been referred to. The full name of PRISMA: Preferred Reporting Items for Systematic Reviews and Meta-Analyses is written in the first sentence of the search strategy paragraph ( where it is first mentioned)

Data Extraction

1. Line 122-123 -: Removing duplicates should be part of study selection and not data extraction.

Authors: Thank you for this valuable feedback on our manuscript. Line 122-123 was moved to study selection and placed in the appropriate position

2. Did the authors use a standardized tool for extraction? If not, was any tool used? If yes, how was the tool developed?

Authors: Thank you for this valuable feedback on our manuscript. We added “We developed and tested an extraction template created using Google Forms.”

Risk of bias assessment and data analysis

1. Can the authors include a justification for the rationale for assessing the quality of the included sources?

Authors: Thank you for raising this point. Assessing the quality of included studies is integral to ensuring the rigor and validity of our review. By evaluating the quality, we can ascertain the reliability of the data and conclusions drawn from these sources.

Results

This section is comprehensive and well-documented.

Areas for Improvement

1. Line 140: meta-analysis. Can the authors justify the use of the word?

Authors: Thank you for this valuable feedback on our manuscript. The use of meta-analysis is an oversight, the word has been replaced with study.

2. Line 165: Is Adamawa the same as Adamaoua (Line 165 and Line 237)?

Authors: Thank you for this valuable feedback on our manuscript. Adamawa is the correct spelling, the correction has been made

3. Line 170-171: Please rework the 2 sentences for improvement in clarity and flow.

Authors: Thank you for this valuable feedback on our manuscript. The two sentences have been edited.

4. Line 176- 217: Please rework this section to capture the interest of the reader. The evidence in this section needs to be synthesized. The authors can consider combining studies with similarities as opposed to listing out the prevalence of MRSA in individual studies. Also, consider the inclusion of a summary statement about the prevalence of MRSA at the beginning of the section e.g., Line 228 and Line 326.

Authors: Thank you for this valuable feedback on our manuscript. The section has been edited.

Discussion

This is comprehensive and well-detailed.

Authors: Thank you

Appendix 1: Registration and Protocol. Can the authors clarify where the registration and protocol information can be located under Methods?

Authors: Our study employed a mapping review approach. Due to the broad objectives associated with mapping reviews, primary registration platforms like PROSPERO do not allow their registration. We edited Appendix 1 accordingly. 

Appendix 2: Include interface for Web of Science

Authors: The Web of Science interface is a web-based platform as the other databases with no specific additional names.

Appendix 6: Can the authors include a description of the headers for the three “settings” to differentiate them from each other?

Authors: Thank you for this valuable feedback on our manuscript. We have organized the data into three distinct categories for clearer understanding: main setting, specific location, and geographical classification.

Reviewer #2: REVIEW OF MANUSCRIPT: PONE-D-23-27108

1. Overall, the manuscript appears to be well-structured, thorough, and provides valuable insights into the Methicillin-Resistant Staphylococcus Aureus (MRSA) situation in Cameroon. Data provided by the researchers is adequate, however, it would be of more benefit if the discussion section provided a more comprehensive analysis of the implications of the research findings.

Authors: Thank you for this valuable feedback on our manuscript.

2. Abstract

The abstract provides a clear and concise overview of the research conducted in this manuscript. It effectively introduces the problem of methicillin-resistant Staphylococcus aureus (MRSA) and its global health threat, with a particular focus on its prevalence, resistance patterns, and genetic characteristics in Cameroon.

a. Even though authors reviewed several papers with information listed in the methods, it is not clear what data was extracted from published work which was included in the study.

Authors: Thank you for this valuable feedback on our manuscript. The necessary adjustment has been made.

b. Line 43: “The genetic diversity of MRSA was significant,” kindly state the level of significance in the study.

Authors: Thank you for this valuable feedback on our manuscript. We edited to use “heterogeneous”.

3. Introduction:

The introduction effectively sets the stage for the research by discussing the global spread of MRSA and its presence in Cameroon, emphasizing the need for surveillance and interventions to curb its dissemination.

a. Include information on global and local prevalence of MRSA if any.

Authors: Thank you for this valuable feedback on our manuscript. The information on global and local prevalence of MRSA has been added in the introduction.

 b. Line 63-66: include reference(s).

Authors: Thank you for this valuable feedback on our manuscript. The references have been added.

c. Line 75-76: include reference(s).

Authors: Thank you for this valuable feedback on our manuscript. The reference has been added.

4. Methods:

The materials and methods section are comprehensive and well-structured. The PRISMA 2020 guidelines, which is a widely recognized approach for systematic reviews establishes the credibility and reliability of the research process. The search strategy, including the databases used and the date of the search, is clearly detailed. The inclusion and exclusion criteria for study selection are well-defined, and the use of the Rayyan systematic review website for study selection is a practical and efficient approach. The data extraction process is thorough, and the risk of bias assessment using the Hoy et al. tool adds rigor to the research.

a. Kindly state how you take care of duplicate papers.

Authors: Thank you for this valuable feedback on our manuscript. As indicated in the study selection section all duplicate articles were removed using EndNote software (version X9, Clarivate Analytics).

b. Kindly represent the bibliographic search strategy in tabular format.

Authors: Added in S2 Table.

c. Line 118-120: Reference flowchart.

Authors: Figure 1 added.

d. Line 132-136: What level of agreement by the authors/investigators was considered during selection process?

Authors: Thank you for this valuable feedback on our manuscript. In our selection process, any disagreements between the two primary reviewers were addressed through thorough discussions as stated. If a consensus could not be reached, a third senior reviewer was consulted to assist in deciding. While we ensured alignment and agreement in our selection process, we did not specifically document or quantify the levels of disagreement that arose. 

5. Result:

The results section is detailed having various subheadings that looked at characteristics of the selected studies, the prevalence of MRSA, antimicrobial resistance patterns of MRSA isolates and the genetic diversity of MRSA in Cameroon.

a. Kindly number the tables/figures appropriately as figures and appendix are used interchangeably. Also, it is important not to include all the information on the studies included in the paper. Key /relevant finding should be included as prose, and this would significantly reduce the word count of the paper.

Authors: Thank you for this valuable feedback on our manuscript. The appendices was edited in Table and Figures et the results section edited and shortened.

b. Line 152: include the setting of the last study.

Authors: We have now provided a detailed description of the prevalence according to the contexts.

c. Kindly provide the summary measure of the various categories of data on the forest plot. E.g., Summary measure of prevalence of MRSA from all the studies included in the paper.

Authors: Thank you for this valuable feedback on our manuscript. We recognize the value of providing such a measure. However, given the substantial heterogeneity in the data with respect to setting, population, and geographical locations, we are cautious about deriving a single pooled value. Such an aggregate might not accurately represent the nuances and variations across the different studies. We aimed to provide a comprehensive overview without oversimplifying the diverse data presented.

6. Discussion:

The discussion section of the manuscript provides a detailed overview of the study's findings regarding the prevalence and characteristics of MRSA in various settings in Cameroon. Overall, it offers valuable insights into the state of MRSA in the region and its implications for public health. However, there are some aspects that could benefit from clarification, refinement, or expansion:

a. Clarity and Organization: The discussion section covers a wide range of topics related to MRSA, including prevalence, antibiotic resistance, genetic diversity, and implications. While comprehensive, it could benefit from better organization and grouping of related information to enhance readability and comprehension. The use of subheadings would be very useful to enhance readability.

Authors: Thank you for this valuable feedback on our manuscript. We edited accordingly. 

b. Limitations:

• Line 540-555: The section mentions several limitations of the study, but it's important to emphasize their impact on the interpretation of the results in regards of generalization and potential biases.

Authors: Thank you for this valuable feedback on our manuscript. We edited accordingly. 

c. Recommendations:

• Line 555-564: The section mentions the need for preventive measures and infection control strategies but could elaborate more on practical recommendations such as healthcare facilities, policymakers, and individuals can take to address the challenges posed by MRSA in Cameroon.

Authors: Thank you for this valuable feedback on our manuscript. We edited accordingly. 

d. Conclusion:

Conclude the discussion section by summarizing the key take aways from the study, the implications for public health in Cameroon, and the broader significance of the findings. This can help readers grasp the main messages more easily.

Authors: Thank you for this valuable feedback on our manuscript. We edited accordingly. 

7. References:

References are relevant to the topic and are recent papers, however 40% of your references are older than 5 years. Kindly include newer references with older references consisting of about 20% of your entire references.

Authors: Thank you for this valuable feedback on our manuscript. The necessary adjustment has been made.

Reviewer #3: Line 27: I suggest if you want to put a significance to Cameroun you may use the bottoms up approach starting with Cameroun and ending with world wide

Authors: Thank you for this valuable feedback on our manuscript. The necessary adjustment has been made.

Line 101: What method this you use for your date? mm/dd/year or dd/mm/year. to prevent such confusion you could write the month out in words

Authors: Thank you for this valuable feedback on our manuscript. The date has been written with the month spelled out in words.

The manuscript is well written and lots of work has been put into it.

I would suggest, as the topic says mapping"... this requires more visual representations like comparing geo-locations,use of charts ,tables etc which would make the work more engaging and easier to grasp as it contains too much information.

Authors: Thank you for this valuable feedback on our manuscript. The Cameroon map with all described data has been added (Figure 3).

Reviewer #4: General comments: The article is generally well written with a good flow. The abstract is a good summary of the work, the objective is clearly stated and the methods clearly describes how to achieve the objectives. The literature review gives enough information based on the objectives and key words. Results are in keeping with the objectives and in in chronologically order. The discussion also provides providing interpretation of the findings in the light of other studies. Limitations for the study are clearly stated and the conclusion is succinct with recommendations clearly state.

Authors: Thank you for this valuable feedback on our manuscript.

Line 42: Will be good to have the range of values for the prevalence in animal, food and environment mentioned as well.

Authors: Thank you for this valuable feedback on our manuscript. The necessary adjustment has been made.

Line 103: For ease of replicating this study, it will be good to specify the key words and medical subject headings used in this study. 

Authors: Thank you for this valuable feedback on our manuscript. The keywords have been added (S2 Table)

Line 150: Will be helpful if you give a brief description of Cameroon in terms of the regions. This will enable the reader appreciate the geographical coverage of the studies. Also good to specify the range of time over which studies were conducted clearly.

Authors: Thank you for this valuable feedback on our manuscript. The study period and A map and a brief description of Cameroon in terms of the regions has been given in the work in the results section under the subtitle “Studies characteristics of the epidemiology of MRSA in Cameroon” 

Line 175: Would have been good to order the prevalence (by year or magnitude) to enable the reader appreciate the trend.

Authors: Thank you for this valuable feedback on our manuscript. We have now organized the prevalence according to settings, populations, and locations.

Line 345: The first 3 sections in the discussion section appear to be a repetition of the results. The discussion section should not be meant for repeating the results.

Authors: Thank you for this valuable feedback on our manuscript. The necessary adjustment has been made.

Line 351 & 373: The abstract and result sections have a prevalence of "1.8 to 46.8%", while the discussion section contains "1.9 to 46.8%". There is need for consistency here.

Authors: Thank you for this valuable feedback on our manuscript. The necessary adjustment has been made (1.9 to 46.8% is correct).

Line 378: Not all the factors mentioned here were well captured in the result section. There is a need to review these sections to make sure the factors referred to are described in the result section.

Authors: Thank you for this valuable feedback on our manuscript. We have now maintained only the factors that are mentioned in the results.

---

## [Decision Letter · Decision Letter 1]

10 Dec 2023

A mapping review of methicillin-resistant Staphylococcus aureus proportions, genetic diversity, and antimicrobial resistance patterns in Cameroon

PONE-D-23-27108R1

Dear Dr. Esemu,

We’re pleased to inform you that your manuscript has been judged scientifically suitable for publication and will be formally accepted for publication once it meets all outstanding technical requirements.

Kind regards,

Mabel Kamweli Aworh, DVM, MPH, PhD. FCVSN

Academic Editor

PLOS ONE

Additional Editor Comments (optional):

Reviewers' comments:

Reviewer's Responses to Questions

**Comments to the Author**

1. If the authors have adequately addressed your comments raised in a previous round of review and you feel that this manuscript is now acceptable for publication, you may indicate that here to bypass the “Comments to the Author” section, enter your conflict of interest statement in the “Confidential to Editor” section, and submit your "Accept" recommendation.

Reviewer #1: All comments have been addressed

Reviewer #3: All comments have been addressed

2. Is the manuscript technically sound, and do the data support the conclusions?

Reviewer #1: Yes

Reviewer #3: Yes

3. Has the statistical analysis been performed appropriately and rigorously? 

Reviewer #1: Yes

Reviewer #3: Yes

4. Have the authors made all data underlying the findings in their manuscript fully available?

Reviewer #1: Yes

Reviewer #3: Yes

5. Is the manuscript presented in an intelligible fashion and written in standard English?

Reviewer #1: Yes

Reviewer #3: Yes

6. Review Comments to the Author

Reviewer #1: Thank you for revising this manuscript and making all major revisions. Please note for future purposes that the protocol for mapping reviews can be registered through other sites including the Open Science Framework.

Minor edits and clarifications

1. Line 117: typo in "previous".

Reviewer #3: All recommendations i gave have been fully included in the manuscript and the work has more quality now

7. PLOS authors have the option to publish the peer review history of their article (what does this mean?). If published, this will include your full peer review and any attached files.

Reviewer #1: No

Reviewer #3: **Yes: **Folashade Onatola Toye

---

## [Editor Report · Acceptance letter]

13 Dec 2023

PONE-D-23-27108R1 

PLOS ONE

Dear Dr. Esemu, 

I'm pleased to inform you that your manuscript has been deemed suitable for publication in PLOS ONE. Congratulations! Your manuscript is now being handed over to our production team.

Kind regards, 

on behalf of

Dr. Mabel Kamweli Aworh 

Academic Editor

PLOS ONE